

# Differential expression of *AtWAKL10* in response to nitric oxide suggests a putative role in biotic and abiotic stress responses

Phearom Bot[1,*], Bong-Gyu Mun[1,*], Qari Muhammad Imran[1], Adil Hussain[2], Sang-Uk Lee[1], Gary Loake[3] and Byung-Wook Yun[1]

[1] Department of Applied Biosciences, Kyungpook National University, Daegu, South Korea
[2] Department of Agriculture, Abdul Wali Khan University Mardan, Mardan, Pakistan
[3] Institute of Molecular Plant Sciences, University of Edinburgh, Edinburgh, UK
* These authors contributed equally to this work.

Corresponding authors
Gary Loake, g.Loake@ed.ac.uk
Byung-Wook Yun,
bwyun@knu.ac.kr

## ABSTRACT

Plant defense against pathogens and abiotic stresses is regulated differentially by communicating signal transduction pathways in which nitric oxide (NO) plays a key role. Here, we show the biological role of *Arabidopsis thaliana* wall-associated kinase (AtWAK) Like10 (*AtWAKL10*) that exhibits greater than a 100-fold change in transcript accumulation in response to the NO donor S-nitroso-L-cysteine (CysNO), identified from high throughput RNA-seq based transcriptome analysis. Loss of *AtWAKL10* function showed a similar phenotype to wild type (WT) with, however, less branching. The growth of *atwakl10* on media supplemented with oxidative or nitrosative stress resulted in differential results with improved growth following treatment with CysNO but reduced growth in response to *S*-nitrosoglutatione (GSNO) and methyl-viologen. Further, *atwakl10* plants exhibited increased susceptibility to virulent *Pseudomonas syringae* pv tomato (*Pst*) DC3000 with a significant increase in pathogen growth and decrease in *PR1* transcript accumulation compared to WT overtime. Similar results were found in response to *Pst* DC3000 avrB, resulting in increased cell death as shown by increased electrolyte leakage in *atwakl10*. Furthermore, *atwakl10* also showed increased reactive oxygen species accumulation following *Pst* DC3000 avrB inoculation. Promoter analysis of *AtWAKL10* showed transcription factor (TF) binding sites for biotic and abiotic stress-related TFs. Further investigation into the role of *AtWAKL10* in abiotic stresses showed that following two weeks water-withholding drought condition most of the *atwakl10* plants got wilted; however, the majority (60%) of these plants recovered following re-watering. In contrast, in response to salinity stress, *atwakl10* showed reduced germination under 150 mM salt stress compared to WT, suggesting that NO-induced *AtWAKL10* differentially regulates different abiotic stresses. Taken together, this study further elucidates the importance of NO-induced changes in gene expression and their role in plant biotic and abiotic stress tolerance.

## INTRODUCTION

Nitric oxide (NO), diatomic molecule is a gaseous free radical known for its signaling role under stress condition (*Delledonne et al., 1998*). This small redox active molecule got fair attention of scientists in the recent past not only due to its signaling but also its regulatory role. Initially identified in mammals with medical roles in cardiac diseases (*Hiraga et al., 1998*; *Liu et al., 2004*; *Fraccarollo et al., 2009*; *Liu et al., 2004*); NO was reported to have equally important roles in plants (*Delledonne et al., 1998*; *Durner, Wendehenne & Klessig, 1998*; *Delledonne, 2005*). In the last couple of decades, NO has been reported to regulate a number of physiological processes such as germination (*Beligni & Lamattina, 2000*) Chlorophyll contents (*Leshem et al., 1997*), adventitious roots development (*Pagnussat et al., 2002*), root elongation (*Gouvêa et al., 1997*), flowering time (*He et al., 2004*) apical dominance (*Lee et al., 2008*) hormonal balance (*Yu et al., 2014*) and plant defense (*Delledonne et al., 1998*; *Feechan et al., 2005*; *Yun et al., 2011*) etc.

Nitric oxide in the presence of oxygen form different important oxides like $NO_2$ that may further react with cellular amines and thiols. It can also react with superoxide anion radical ($O_2^-$) producing such ions that later on cause significant damage to cell structure (*Wendehenne et al., 2001*). These molecules are also called reactive nitrogen species (RNS). The production of reactive oxygen and nitrogen species (ROS and RNS) is an important outcome in response to biotic and abiotic stresses (*Garcia-Mata & Lamattina, 2002*; *Burniston & Wilson, 2008*). Controlled ROS production is extremely important for activating downstream processes. For example, hydrogen peroxide ($H_2O_2$) leads to modulation of the cellular redox state and may regulate certain defense-related processes including programed cell death (PCD) and biotic (*Harding et al., 2003*; *Suzuki et al., 2011*) and abiotic stress responses (*Madhava Rao & Sresty, 2000*). However, contrary to well-studied ROS, little is known about the RNS and their roles in plants.

Plants have evolved fine-tuned mechanisms to cope with attempted pathogen ingress. One of these defense responses includes the evolution of resistance (*R*) genes to defend themselves against pathogen attacks. *R*-genes encode proteins that either recognize pathogen avirulence (*Avr*) proteins or their activities (*Dangl & Jones, 2001*; *Jones & Dangl, 2006*). Pathogen recognition prevents pathogen expansion via a variety of mechanisms, including localized PCD which may involve signaling components analogous to animal apoptosis (*Plocik, Layden & Kesseli, 2004*).

The intracellular secondary messenger guanosine 3′, 5′-cyclic monophosphate (cGMP) has been reported to be an important signaling molecule controlling a wide range of physio-molecular responses in both prokaryotes and eukaryotes (*Schaap, 2005*). In higher plants cGMP is reported to be involved in signaling during various physiological processes (*Newton & Smith, 2004*). These include biotic (*Durner, Wendehenne & Klessig, 1998*), abiotic (*Pasqualini et al., 2009*) and NO-dependent signaling (*Prado, Porterfield & Feijo, 2004*). After the perception of stress responses, the guanylyl cyclase (GC) activates to catalyze the synthesis of cGMP from guanosine 5-triphosphate (GTP) (*Schaap, 2005*). However, despite the well-explored role of cGMP in signaling, the role of its catalytic enzyme GCs in higher plants is yet to be identified. BLAST searches have identified about

seven candidates GCs in Arabidopsis (*Meier et al., 2010*). One of these candidates is the Arabidopsis wall-associated kinase-like 10 ((AtWAKL10) AT1G79680). The WAK/WAKL genes are typically supposed to encode a class of receptor-like protein kinases having a transmembrane domain and a serine/threonine kinase domain in addition to an extracellular region that is closely associated with the cell wall (*He et al., 1999*). Reports suggested that some WAKs may be involved in pathogen defense responses. For example, *AtWAK1* expression was induced after *Pseudomonas syringae* infection in Arabidopsis in *NON-EXPRESSOR OF PR1 GENE* (*NPR1*) dependent manner. Furthermore, *AtWAK1* is also reported to be involved in systemic acquired resistance (SAR) (*Maleck et al., 2000*).

Nitric oxide transfers its bioactivity predominantly through a key post-translational modification termed *S*-nitrosylation: a cysteine-based protein modification in which a NO moiety is covalently attached to a solvent-exposed cysteine residue to form an *S*-nitrosothiol (*Stamler et al., 1992*; *Khani et al., 2017*). These SNOs then interact with intracellular sulfhydryl-containing molecules and are of great importance as they are more stable than NO (*Leterrier et al., 2011*). Among various SNOs, *S*-nitrosoglutathione (GSNO) is produced by the *S*-nitrosylation reaction of both NO with glutathione (GSH) and bears special importance since it is reported to be a mobile reservoir of NO (*Stamler, Lamas & Fang, 2001*). Cellular GSNO homeostasis is controlled by the enzyme GSNO reductase (GSNOR) (*Liu et al., 2001*) which is functionally conserved across the animal, plant and bacterial species. Research studies involving GSNOR support its ameliorating role during GSNO-mediated nitrosative stress (*Foster et al., 2009*; *Stamler, Lamas & Fang, 2001*). These properties make GSNOR vital enzyme for plant growth as cellular redox status is crucial to plant growth, development and environmental interactions (*Malik et al., 2011*). The loss-of-function mutation in GSNOR termed as *atgsnor1-3* resulted in a significant reduction in plant growth and compromised defense response in *Arabidopsis thaliana* (*Feechan et al., 2005*; *Yun et al., 2011*).

Microarray and RNA-seq analysis have revealed a number of key genes showing differential expression to NO or its donors (*Parani et al., 2004*; *Begara-Morales et al., 2014*). Recently, *Hussain et al. (2016)* demonstrated several hundred genes showing differential response to the NO donor S-nitroso-L-cysteine (CysNO). Similarly, 673 transcription factors (TFs) including major TF families showed differential expression to the same NO donor (*Imran et al., 2018a*). Here we show *AtWAKL10* a candidate gene of NO signaling that exhibited more than a 100-fold induction (Fold change 124.7) in response to CysNO and exhibits a role in both plant defense and oxidative stress for this gene product. *AtWAKL10* is a positive regulator of basal defense, effector-triggered immunity and salt stress while negatively regulating drought stress.

# MATERIALS AND METHODS

## Plant material, growth conditions and genotyping

Col-0 was used as wild type (WT), whereas *atgsnor1-3*, a loss-of-function mutant of Arabidopsis *S*-nitrosoglutathione reductase1 (*AtGSNOR1*) was used as a susceptible control (*Feechan et al., 2005*). Loss-of-function mutant lines for *WAKL10* (AT1G79680) was procured from the Arabidopsis Biological Resource Center (ABRC, https://abrc.osu.edu/).

Seeds were surface-sterilized using 50% bleach (commercial bleach) for about 1~2 minutes and grown on half-strength Murashige and Skoog (MS) media (*Murashige & Skoog, 1962*). After a week, seedlings were transplanted in soil and grown under controlled light (16–8 h.) and temperature (23 °C ± 2). *atwakl10* plants were genotyped to identify homozygous lines using gene-specific forward and reverse and T-DNA border primers using PCR. The list of primers is given in Table 1. The PCR conditions were initial denaturation at 94 °C for 2 min followed by 30 cycles of 94 °C for 20 s, 58 °C for 30 s and 72 °C for 1 min and a final extension at 72 °C for 5 min. The identified homozygous lines were then used for further experiments.

## Oxidative and nitrosative stress assay

To study the response of *atwakl10* towards oxidative and nitrosative stresses plants were subjected to oxidative and nitrosative stress as described earlier (*Imran et al., 2016*). Briefly, for nitrosative stress conditions, sterilized Arabidopsis seeds from WT Col-0, *atgsnor1-3* and *atwakl10* were sown on media supplemented with one mM CysNO or one mM GSNO. For exposure to oxidative stress, sterilized seeds from Col-0 and mutant line were grown on media supplemented with two mM $H_2O_2$ or one μM Methyl Viologen. For all treatments, plants were grown in 16 and 8 h light and dark conditions, respectively in triplicates at 23 °C ± 2 temperature. The experiments were repeated twice and cotyledon development frequency (CDF) was recorded after 1 week as described by *Imran et al. (2016)*. The term CDF was used for green developed cotyledons.

## Pathogen inoculations and cell death

*Pseudomonas syringae* pv. *tomato* (*Pst*) strain DC3000 was cultured and maintained as described (*Whalen et al., 1991*). Briefly, virulent *Pst* DC30000 (concentration 0.0002 at $OD_{600}$ i.e. $5 \times 10^5$ CFU mL$^{-1}$) and avirulent *Pst* DC30000 *avrB* (concentration 0.002 at $OD_{600}$ i.e. $1 \times 10^6$ CFU mL$^{-1}$) were grown on LB (Luria-Bertani) media with appropriate antibiotics (Rifampicin, Kanamycin and Rifampicin, respectively). The single colony was then transferred to LB broth and incubated at 28 °C overnight with shaking. Both the bacterial strains were then harvested by centrifugation in 10 mM MgCl$_2$. The inoculum thus prepared was infiltrated into the abaxial side of leaves at the indicated concentration. After 2 and 4 days of inoculation, disease symptoms were recorded and leaf samples (four replicates) were harvested for bacterial growth analysis (*Feechan et al., 2005*).

## Histological staining

The avirulent strain of *Pst* DC3000 expressing the *avrB* effector protein interacts with plant R-genes thereby triggering hypersensitive cell death response (HR) (*Leister, Ausubel & Katagiri, 1996*). To see pathogen-induced cell death in vivo, all the genotypes were infiltrated with *Pst* DC3000 avrB and cell death in inoculated leaves was measured by trypan blue staining (*Yun et al., 2011*). To visualize $H_2O_2$ in situ we used the already established method (*Yun et al., 2003*). Briefly, leaves inoculated with *Pst* DC3000 *avrB* were stained with 3-3′-diaminobenzidine (DAB) which is absorbed by the plant and polymerized locally in the presence of $H_2O_2$ and peroxidases giving a visible brown stain.

**Table 1 List of primers used.**

| Gene name | Acc No | Forward (5′-3′) | Reverse (5′-3′) |
|---|---|---|---|
| Actin | AT3G18780 | GCTGGACGTGACCTTACTGA | CCATCTCCTGCTCGTAGTCA |
| PR1 | AT2G14610 | GTGCAATGGAGTTTGTGGTC | TCACATAATTCCCACGAGGA |
| AtWAKL10 | AT1G79680 | CACTAGCGCATGAACATGTTG | AAACCCGTCTCTGCTTTTAGC |
| Sail border primer | SAIL_LB1 | GCCTTTTCAGAAATGGATAAATAGCCTTGCTTCC | |

The inoculated leaves were dipped in DAB solution and put on a shaker overnight. The DAB solution was then replaced with 96% ethanol and boiled for 10 min. The leaves were then allowed to cool-down and washed with ddH$_2$O two times. The leaves were then mounted on microscopic slides.

## Electrolyte leakage

Pathogen-induced cell death was quantified by measuring electrolyte leakage. Briefly, one cm$^2$ leaf discs form the leaves inoculated with avirulent *Pst* DC3000 at a concentration of $1 \times 10^8$ CFU mL$^{-1}$ were collected in triplicates with three leaf discs per replicate. The leaf discs were washed with deionized water and put in a multi-well plate (SPL life sciences, Pocheon-si, Korea) containing an equal amount of deionized water. Electrolyte leakage in each well was measured over time by conductivity meter (Horiba Twin Cond, Conductivity meter B-173).

## Salt and drought stress

To test *atwakl10* response to salt stress, seeds were placed on ½ MS media in petri dishes supplemented with 100 and 150 mM salt (NaCl). The control plants were grown on ½ MS media only. The petri plates were placed in the growth chamber at 23 °C ± 2 temperature for 16 and 8 h of light and dark period respectively. One week later, CDF was measured and pictures were taken. The drought stress was induced by withholding water for two weeks. The control plants were watered regularly. After 2 weeks, plants were examined and re-watered to see the recovery. The recovery rate was calculated as the number of recovered plants after re-watering to the number of total plants subjected to drought stress for each genotype.

## Quantitative real-time (qRT) PCR

RNA extraction and qRT-PCR were performed as mentioned earlier. Briefly, total RNA was extracted from fresh leaf tissues using the Qiagen RNeasy Plant mini kit (Qiagen, Seoul, Korea). After treatment with DNase I, one μg of total RNA was used for the synthesis of the first strand of cDNA using Omniscript RT kit and oligo dT. The cDNA was then used as a template for running RT-PCR. Real-time PCR was performed in 48-well plate with Eco Real-time PCR System (Eco$^{TM}$ Illumina, San Diego, CA , USA). In each reaction, 400 ng of the template cDNA, 100 nM of forward primer and reverse primer, 10 μl of SYBR green master mix was used in a total of 20 μl reaction volume. *Pathogenesis-related gene 1* (*PR1*), expression was studied relative to actin expression. The primer list is given in Table 1.

## Promoter analysis of *AtWAKL10*

Promoter sequence 1.5 kb upstream of the transcription initiation site was analyzed using PlantPAN 2.0 (*Chow et al., 2015*). Briefly, accession number of *AtWAKL10* was used to search *cis*-regulatory elements in the promoter using *Arabidopsis thaliana* weight matrix. The upstream and downstream coordinates of promoter region were X: 1500, Y: 100, Z: 500. The Arabidopsis and Rice were selected as an option for identifying conserved regions. After execution of the analysis, TFs and their binding sites within the promoter region were visualized by selecting a few key TFs. Furthermore, the conserved regions among the promoter of *AtWAKL10* and its orthologs in another model plant rice were studied using Cross-species tool of PlantPAN.

## Statistical analysis

For all the experiments, the data were analyzed for significance and *p*-value was determined using a two-tailed student's *t*-test using Microsoft Excel. The level of significance was set at $p \leq 0.05$ or $p \leq 0.01$. The mean and standard error were calculated and compared with control. All the experiments were repeated from two to four times and representative results are presented.

# RESULTS

## Loss of *AtWAKL10* function results in differential responses to oxidative and nitrosative stress

T-DNA insertion in *atwakl10* was genotyped to identify homozygous lines and further confirmed by examining its expression level in WT and *atwakl10* plants. No *AtWAKL10* expression was detected in the mutant line (Fig. 1A; Fig. S1). Cotyledon development frequency was used as a score for plant fitness. Our results suggested that the loss-of-function mutant, *atwakl10*, showed significantly higher CDF compared to WT under CysNO-induced nitrosative stress, however, in case of GSNO, there was no significant difference (Figs. 1B and 1C). In contrast, on MV media there was no significant difference in CDF among WT and *atwakl10* plants (Fig. 1C). Interestingly, *atgsnor1-3* showed significant tolerance towards MV stress (Fig. 1C). Together all these data suggest that *AtWAKL10* differentially regulate different stress conditions. Under normal conditions, *atwakl10* plants were phenotypically similar to WT in height however, *atwakl10* showed fewer branches compared to WT (Fig. 1D).

## *AtWAKL10* is required for basal defense

To validate the possible role of *ATWAKL10* in basal defense, mutant lines, as well as WT were inoculated with virulent *Pst* DC3000. The *atgsnor1-3* is susceptible to virulent *Pst* DC3000 compared to WT (*Feechan et al., 2005*; *Yun et al., 2011*). Symptoms of pathogen infection were clearly visible on infiltrated leaves in both *atgnor1-3* and *atwakl10* lines (Fig. 2A). The symptoms development results were also reflected in pathogenicity assay. We observed that at 0 day post inoculation (dpi), only *atgsnor1-3* showed significant ($p \leq 0.001$) increase in pathogen growth while no significant difference was found in *atwakl10* mutants compared to WT (Fig. 2B). However, both *atwakl10* and *atgsnor1-3*

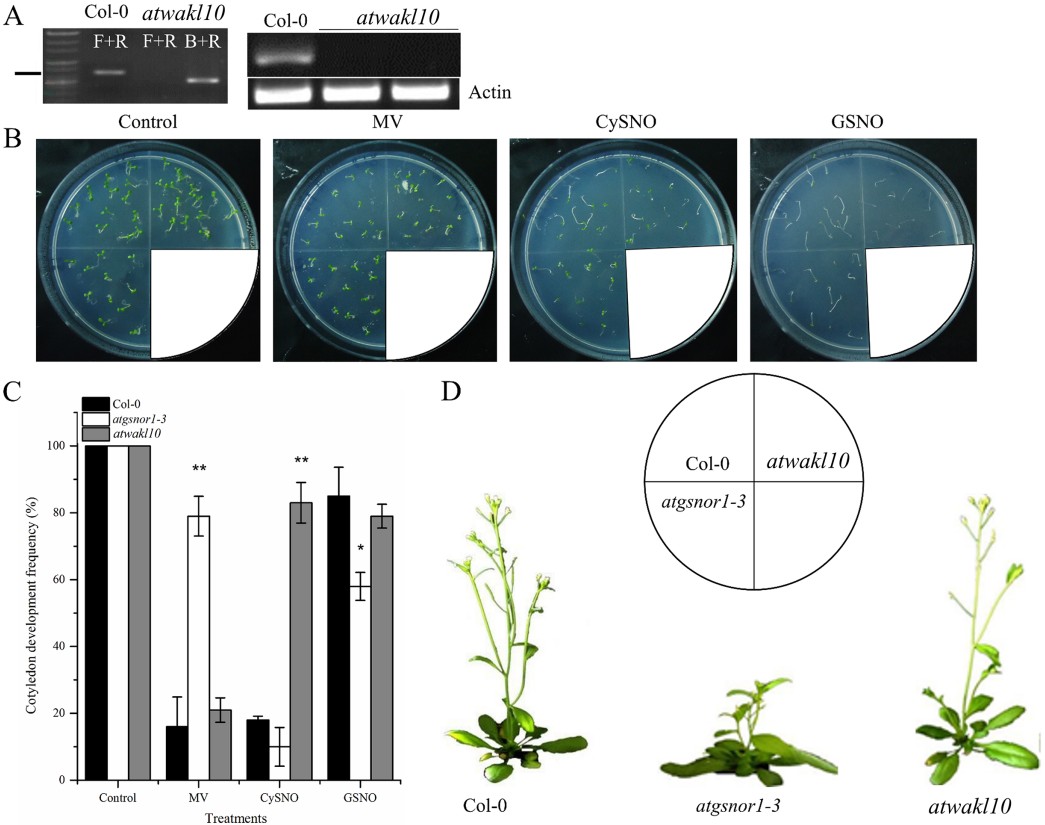

**Figure 1 The response of *atwakl10* plants to oxidative and nitrosative stress.** (A) Genotyping and expression of *AtWAKL10* in WT and *atwakl10* mutants. (B) Germination of *atwalk10* and WT plants on control, oxidative (MV induced) and nitrosative stress (CysNO and GSNO induced) media. (C) CDF of *atwakl10* plants under indicated stressed conditions. (D) Phenotypes of the indicated genotypes. All the data points in (C) are the mean of three replicates and the experiment was repeated about two times with similar results. The labels F+R in (A) denotes gene-specific forward and reverse primers while B denotes Sail left border primer. Error bars represent ± SE. Significant differences are represented by asterisks (Student's *t*-test with 95 (*) and 99% (**) confidence levels.

mutant showed significant increase in bacterial growth compared to WT at 2 and 4 dpi (Fig. 2B). We hypothesized that the increased pathogen susceptibility observed in the *atwakl10* line may be due to down-regulation of the salicylic acid (SA) signaling pathway, therefore, we tested *Pathogenesis-related 1* (*PR1*) gene expression which is a marker for SA-dependent gene expression. *PR1* showed significantly reduced ($p \leq 0.05$) transcript accumulation in both *atgsnor1-3* and *atwakl10* plants overtime compared to WT (Fig. 2C). However, at 0 h post inoculation, *atwakl10* showed increased *PR* accumulation. This suggests that the susceptible phenotype of *atwakl10* may be due to down-regulation of SA-dependent *PR* genes.

### *AtWAKL10* positively regulates effector-triggered immunity

To study a possible role of AtWAKL10 in effector-triggered immunity (ETI), plants were infiltrated with avirulent *Pst*DC3000 expressing *avrB*. A key defense response to restrict

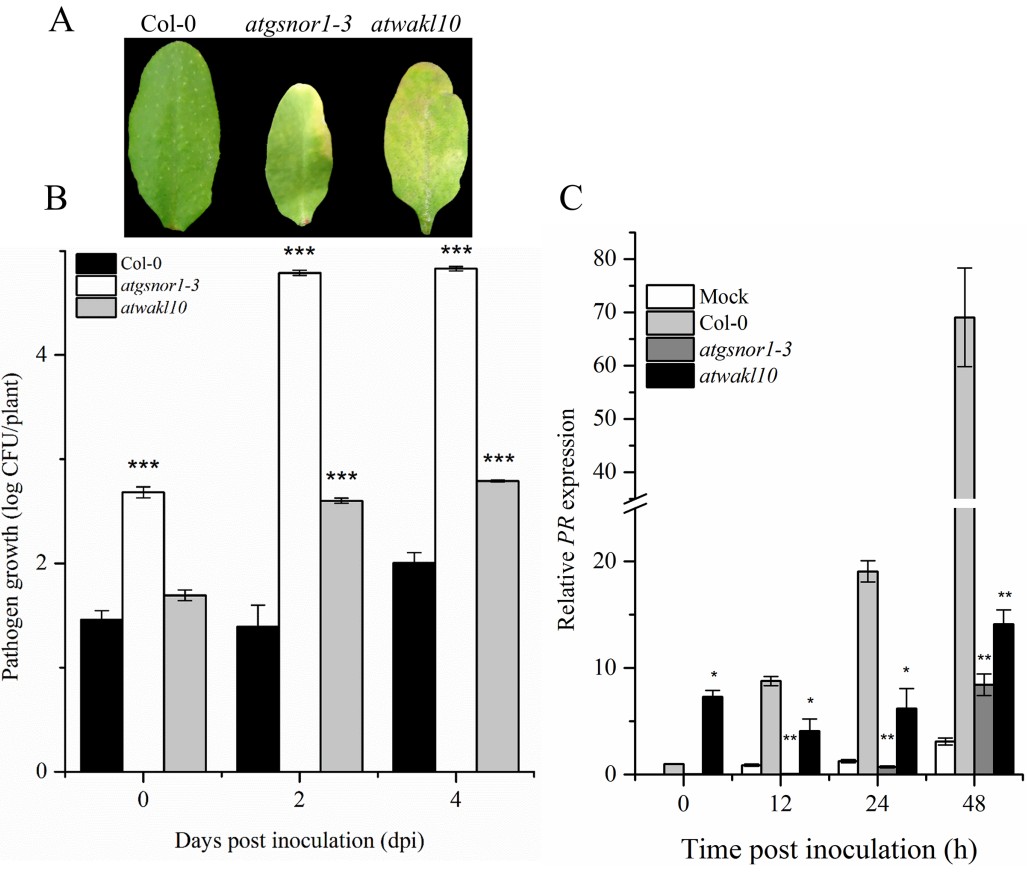

**Figure 2** *AtWAKL10* **negatively regulates basal defense.** (A) Symptom development after virulent *Pst* DC3000 inoculation. (B) Pathogen growth after virulent *Pst* DC3000 inoculation and (C) *PR1* gene relative expression after attempted *Pst* DC3000 (virulent) inoculation. The data points are the mean of at least three replicates. The pathogenicity experiment was repeated four times with similar results. Error bars represent ±SE while asterisk represents significant difference compared to WT (Student's *t* test). The asterisk * represents $p \leq 0.05$, ** represents $p \leq 0.01$ while *** represents $p \leq 0.001$.

pathogen growth in resistant plants is the hypersensitive response (HR). Therefore, we sought to determine the pathogen-induced HR response in both WT and *atwakl10* mutant plants using trypan blue staining (*Feechan et al., 2005*). Our results suggested an increased cell death phenotype in both *atwakl10* and *atgsnor1-3* plants compared to WT (Fig. 3A). Similarly, pathogen growth assay showed a significant ($p \leq 0.05$ and $p \leq 0.001$ respectively) increase in number of bacterial colonies in both *atwakl10* and *atgsnor1-3* plants compared to WT plants at 2 and 4 days after inoculation (Fig. 3B); though, there was no significant difference in bacterial growth at 0 dpi among all genotypes. We further suggested that the increased HR response in *atwakl10* plants might be due to increased ROS accumulation. Therefore, we studied $H_2O_2$ accumulation after *Pst* DC3000 *avrB* inoculation using DAB staining. Our results suggested increased $H_2O_2$ accumulation in *atwakl10* compared to WT (Fig. 3C). Hypersensitive response is often characterized by cell death at the site of infection. Therefore to further confirm the pathogen-induced cell death phenotype of *atwakl10*, electrolyte leakage was measured overtime following

*Pst*DC3000 *avrB* challenge. The *atwakl10* plants showed more electrolyte leakage overtime compared to other genotypes studied; followed by *atgsnor1*-3 (Fig. 3D).

## Promoter of *AtWAKL10* rich in binding sites for stress response transcriptional regulators

The gene structure is crucial to study the function of a gene and plays a key role in adaptation and evolution. We, therefore, analyzed the gene structure of *AtWAKL10* and found that it consists of three exons and two intronic regions (Fig. 4A). Promoter region upstream of transcription initiation site regulates mechanistic control of transcription initiation by possessing sites for binding TFs (*Palmieri et al., 2008*; *Imran et al., 2018a*). These sites also called *cis*-regulatory elements regulate several cellular processes. We, therefore, analyzed the promoter of *AtWAKL10* 1.5 kb upstream of the transcription initiation site to identify *cis*-regulatory elements. A number of key *cis*-regulatory elements were identified in the promoter region (complete list given as Table S1). Some of them were mentioned in Fig. 4B including binding sites for *Arabidopsis thaliana MERISTEM LAYER1* (*ATML1*) a TF which is expressed in the L1 layer and is supposed to suggest the mechanism for cell fate specification (*Abe, Takahashi & Komeda, 2001*). Among others were *AtDREB2* involved in regulation of dehydration and high salinity responsive gene expression (*Nakashima et al., 2000*) and *AtDREB19* that enhanced plant performance without compromising phenotype under drought and salt stress (*Krishnaswamy et al., 2011*). Furthermore, the important w-boxes, binding sites for WRKY TFs were also present (Fig. 4B). Identification of *AtWAKL10* in model crop rice showed that AtWAKL10 is similar to rice OsWAKL5 (Fig. 4C).

## *AtWAKL10* positively regulates NaCl-induced salt stress

The promoter region of *AtWAKL10* showed TF binding sites (TFBS) for TFs that are involved in abiotic stress responses as mentioned in the above section. We, therefore, tested if there is any possible role for AtWAKL10 in abiotic stress. Wild type and *atwakl10* lines were subjected to 100 and 150 mM NaCl solution and the germination percentage was recorded after 1 week. Results suggested that at 100 mM salt solution there was no significant difference in germination percentage of *atwakl10* and WT plants; whereas, germination was significantly reduced at 150 mM salt treatment (from 100% in control media to 54% in 150 mM) Figs. 5A and 5B)).

## *atwakl10* plants exhibit increased drought tolerance

Functional characterization of downstream NO-regulated genes revealed that they play significant roles in drought stress responses (*Shi et al., 2014*). Furthermore, the promoter of *AtWAKL10* was having TFBS for *DREB19* and *DREB2*. *DREB2* is reportedly involved in drought tolerance (*Liu et al., 1998*). We, therefore, investigated the response of *atwakl10* plants to drought stress. Thus, WT and *atwakl10* lines were exposed to drought through water-withholding for 2 weeks (Figs. 6A and 6B). Our results indicated that *atwakl10* plants exhibited improved drought tolerance compared to WT (Fig. 6C). Most of the *atwakl10* plants exhibited reduced wilting and recovered quickly after re-watering.

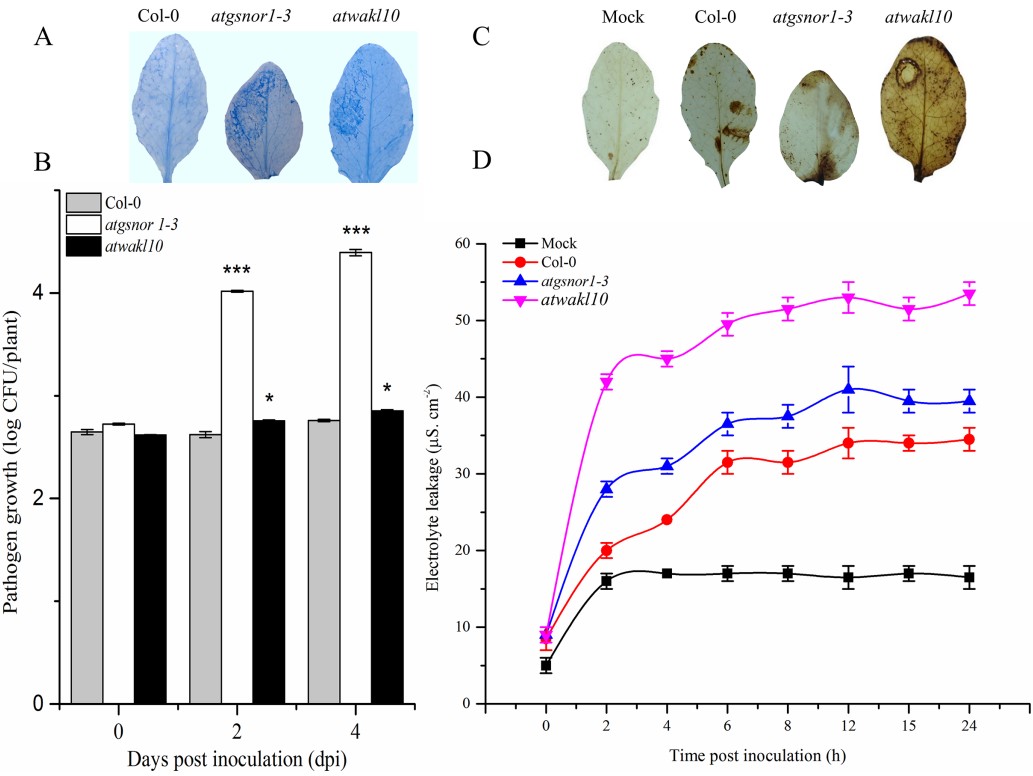

**Figure 3 AtWAKL10 positively regulates effector-triggered immunity.** (A) Trypan blue staining showing HR response. (B) Pathogen growth determined as log colony forming unit (CFU) (C) Diaminobenzidine (DAB) staining depicting $H_2O_2$ accumulation and (D) cell death-induced electrolyte leakage after *Pst* DC3000 *avrB* challenge. The background in A and C are modified for more clarity. The data points in B and D are the mean of three replicates. The pathogenicity experiment was repeated about three times while that of electrolyte leakage was repeated twice with similar results. Error bars represent ± SE ($n = 3$) while significant differences compared to WT are marked with asterisks (Student's *t* test). The asterisk * represents $p \leq 0.05$, ** represents $p \leq 0.01$ while *** represents $p \leq 0.001$.

In contrast, few WT plants survived the drought treatment. The recovery rate for *atwakl10* plants was the highest (60%) compared to WT (0%) (Fig. 6D).

## DISCUSSION

Most biotic and abiotic stresses have a common feature: the production of oxidative and nitrosative burst and an associated change in cellular redox potential. Exposure to RNI and ROS leads to alterations in plant redox homeostasis and can even result in cell death. Therefore, it is of paramount importance to explore the role of NO in the regulation of genes involved in key plant processes. Recently, we have reported more than 6,000 genes that showed significant differential expression in response to the NO donor CysNO (*Hussain et al., 2016*). Among those, *AtWAKL10* was one candidate gene that showed more than 124-fold induction suggesting *AtWAKL10* might be linked to NO signaling during biotic and/or abiotic stress conditions. The present investigation strongly supports a role for NO in the regulation of plant defense responses to biotic and abiotic factors. *WAKL10* encodes a guanyl cyclase, an enzyme that catalyzes the synthesis of

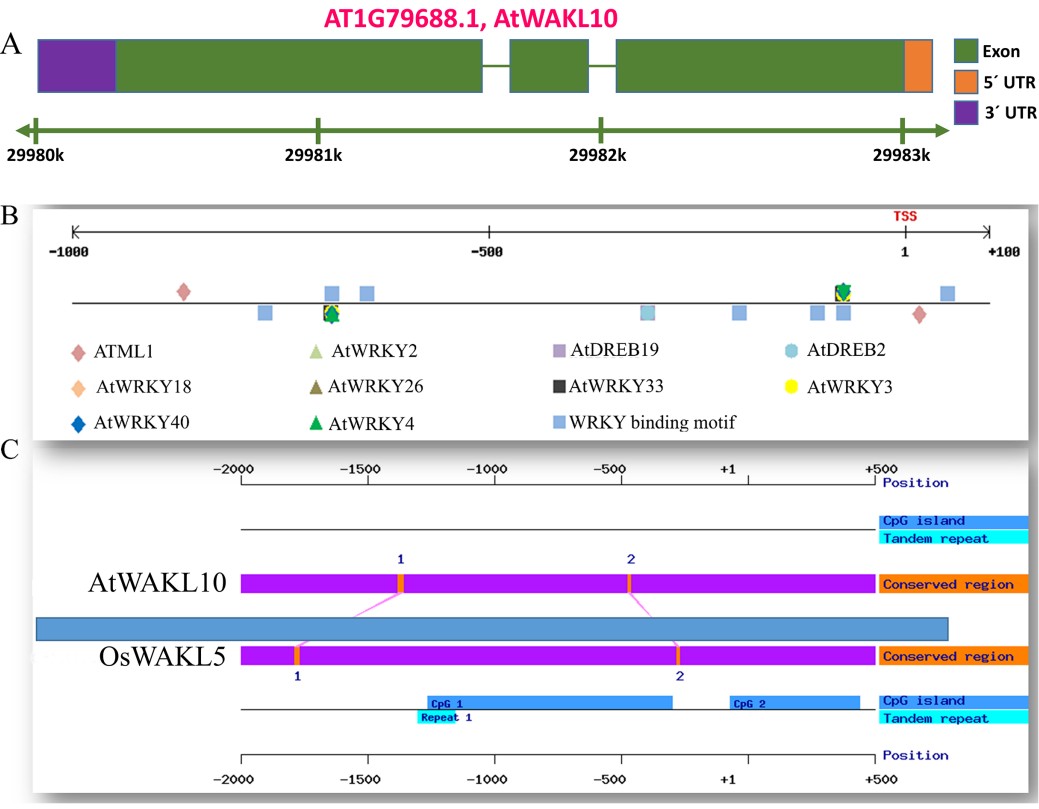

**Figure 4 Promoter analysis of *AtWAKL10*.** (A) Genetic structure of *AtWAKL10*. Green bars represent exons, while gaps between two exonic regions are introns. (B) Position and visualization of different transcription factor binding site (TFBS) one Kb upstream of transcription initiation site and (C) comparison of conserved regions of *AtWAKL10* and its orthologs in rice.

cGMP from GTP (*Schaap, 2005*). In animals, cGMP stimulates biosynthesis of cyclic ADP-ribose and thereby serves as a further downstream messenger of NO. Guanosine 3′, 5′-cyclic monophosphate levels are modulated by NO in animal cells and equilibrium concentrations of cGMP are dependent on NO-activated guanylate cyclases (*Ignarro, 2000*). Our results suggested that *atwakl10* plants showed enhanced tolerance to the NO donor CysNO but not GSNO (Fig. 1C). However, on both NO donors-mediated nitrosative stress conditions, the *atwakl10* plants showed reduced growth and albino phenotype with more severe on GSNO-mediated nitrosative stress compared to WT. This might be because of the possible role of *AtWAKL10* in RNS scavenging which may be due to its many fold induction (>124) in response to CysNO. Therefore a perturbation in the function may result more nitrosative stress which ultimately results in compromised plant growth as suggested by many reports (*Corpas & Palma, 2018*; *Lee et al., 2008*; *Lindermayr, 2017*). The difference in response of *atwakl10* plants to different NO donors might be because of the differential accumulation of intracellular NO due to different NO release rates of NO donors (*He & Frost, 2016*). However, GSNO being biologically active molecule present in the cell that serves a stable reservoir of NO having the potential to modify different proteins (*Frungillo et al., 2014*). Further detail investigation is required to

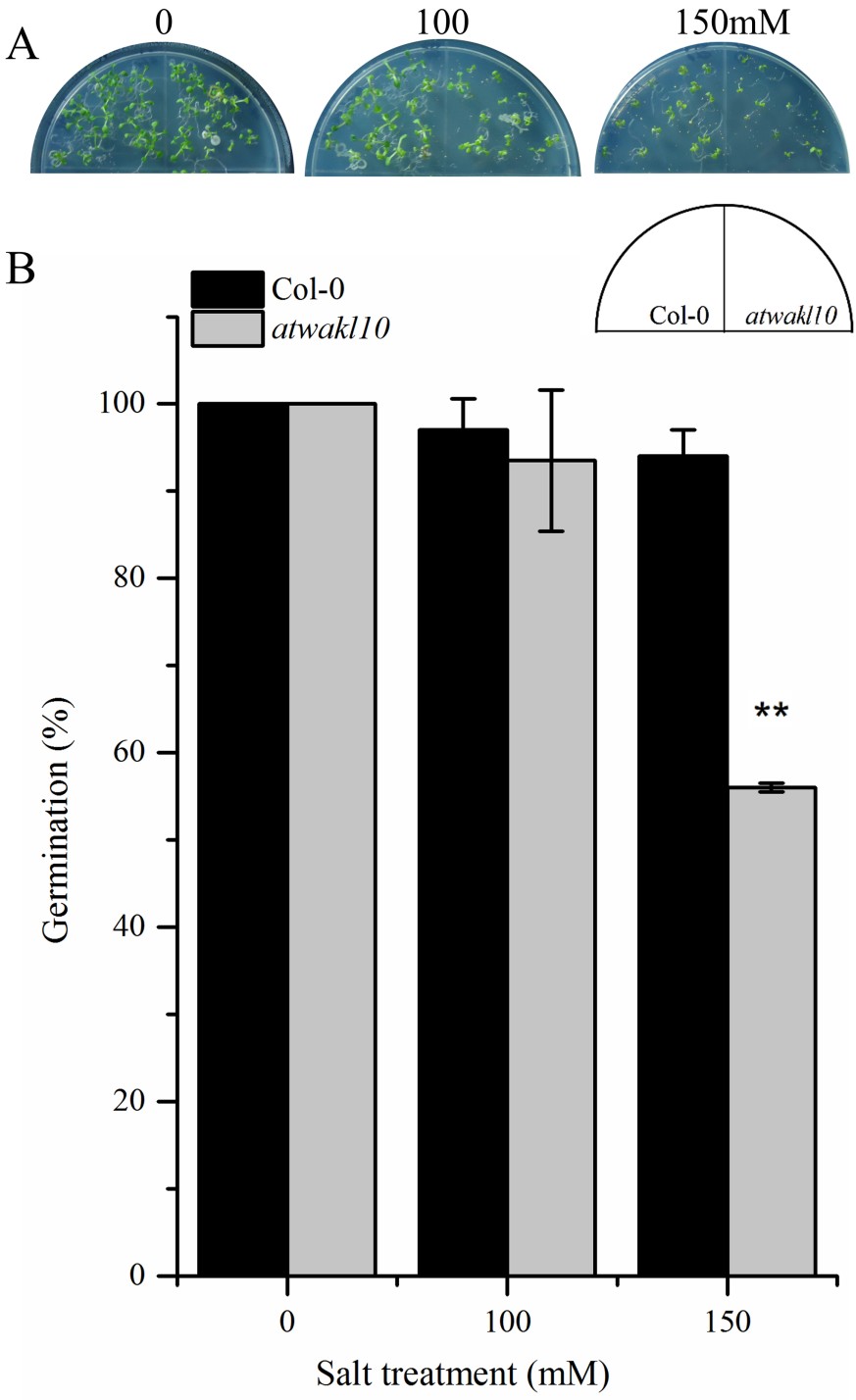

**Figure 5 *atwakl10* plants showed reduce germination percentage under high salinity conditions.**
(A) Germination of *atwakl10* and WT plants on different salt conditions. (B) Germination percentage in WT and mutant line after exposure to different salinity conditions. The data in the graph represents the mean of three replicates. The experiment was repeated three times with similar results. Error bars represent ± SE (*n* = 3). (Student's *t* test). The asterisk * represents $p \leq 0.05$ while ** represents $p \leq 0.01$.

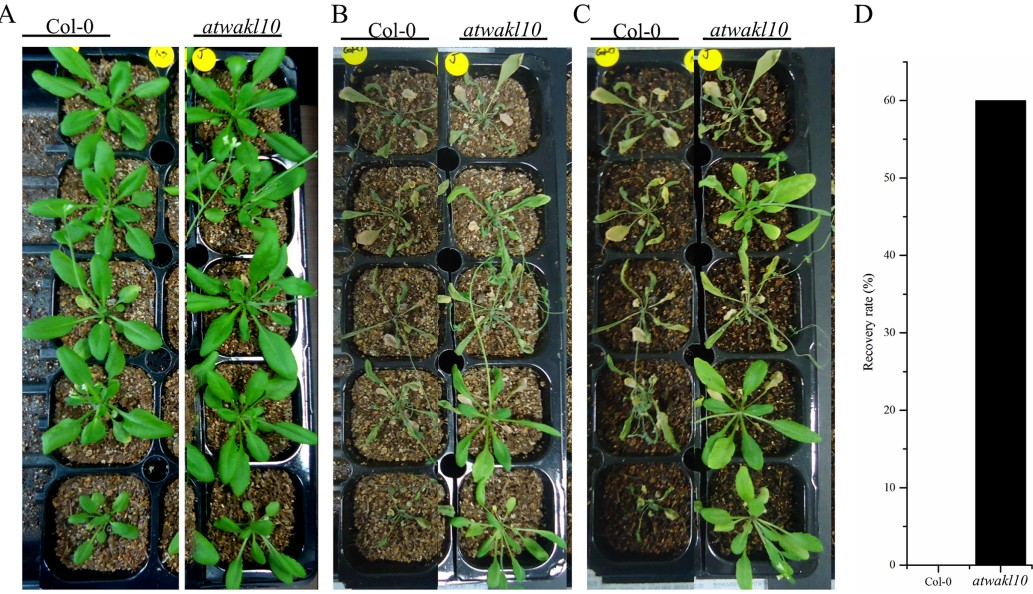

**Figure 6** *atwakl10* **plants exhibit drought tolerance.** (A) Phenotypes of indicated genotypes 0 days post-drought (after withholding water). (B) After 2 weeks of drought stress. (C) Phenotypes of plants after 2 weeks of drought stress, followed by re-watering. (D) Quantification of recovery rates after re-watering. The experiment was repeated twice with similar results.

unravel the underlying mechanism of how NO regulates growth responses during nitrosative stress condition. On MV-induced oxidative stress media, both WT and *atwakl10* showed reduced CDF while *atgsnor1-3* showed increased tolerance with about 80% CDF percentage. The enhanced growth performance by *atgsnor1-3* on MV media might be due to a strong antioxidant system in *atgsnor1-3* (*Kovacs et al., 2016*).

We further studied the role of AtWAKL10 in basal defense and found that *atwakl10* plants were more susceptible to virulent *Pst* DC3000, suggesting that *AtWAKL10* positively regulates basal defense (Figs. 2A and 2B). Previous studies on *AtWAK* genes have suggested their possible role in pathogen defense responses. In Arabidopsis, *AtWAK1* was induced by *P. syringae* as well as by exogenous SA application (*He, He & Kohorn, 1998*). In addition, *AtWAK1* transcription is up-regulated by SAR inducing conditions (*Maleck et al., 2000*).

WAK/WAKL proteins comprise a transmembrane domain, a cytoplasmic serine/threonine kinases domain and an extracellular calcium-binding domain (*Meier et al., 2010*). The intracellular messenger cGMP is an important signaling molecule in biotic stress responses in plants. *Ludidi & Gehring (2003)* showed in vitro generation of cGMP from a GC motif present in an intracellular domain of the AtWAKL10 protein. WAKL has a well-documented role in regulating SA-dependent defense response (*He, He & Kohorn, 1998*). Few WAKL genes have been implicated in SA/JA cross-talk. The susceptibility of *atwakl10* to *Pst* DC3000 has indicated the role of *WAKL10* in defense signaling (Figs. 2A and 2B). Expression dynamics of SA responsive genes have provided evidence for the involvement of *WAKL10* gene in SA responses. Further, the promoter region of

Arabidopsis *AtWAK* and *AtWAKL* genes harbor binding sites for WRKY TFs (*Meier et al., 2010*). These reports are in accordance with our results in which we also found WRKY TFBS in the promoter region of *AtWAKL10*. This TF family has been extensively studied for its role in regulating the expression of defense response genes (*Eulgem et al., 1999*). *Li, Brader & Palva (2004)* described the role of Arabidopsis WRKY70 as a node of convergence between SA and JA signaling. They reported overexpression of *WRKY70* caused activation of SA-induced *PR* genes and suppression of JA responsive *PDF1.2* gene (*Meier et al., 2010*). Recently we have reported a member of the WRKY TF family that was regulated at the transcriptional level in response to NO, in an RNA-seq mediated transcriptome analysis (*Imran et al., 2018b*). To study the role of AtWAKL10 in ETI, we inoculated plants with *Pst* DC3000 expressing *avrB* effector. The ultimate response of effector-triggered immunity as a result of interaction between *avr* and *R* gene products is HR (*Jones & Dangl, 2006*). Though the exact mechanisms of *R-avr* interactions are still being investigated, the requirement of SA has been shown in many plant-pathogen interactions (*Mauch-Mani & Slusarenko, 1996*). Therefore, we quantified the HR response by histological staining using trypan blue. We found that *atwakl10* lines showed an enhanced HR response (Fig. 3A). The HR is mostly associated with resistance to Biotrophic pathogens; however, *atwakl10* plants showed a significantly greater bacterial titer compared to WT (Figs. 2B and 3B). Similarly, the *atgsnor1-3* mutant line also showed increased susceptibility to virulent and avirulent *Pst* DC3000 with increased CFU counts (Figs. 2B and 3B). These results are in accordance with *Feechan et al. (2005)* and who for the first time reported that perturbation in *GSNOR1* function results in a compromised defense response. One of the possible explanations for this could be the hemibiotrophic nature of *Pst* which even can feed and survive on dead matter and multiply as suggested by (*Lee & Rose, 2010*). Second, after attempted pathogen infection, the production of ROS such as superoxide ($O_2^-$) and $H_2O_2$ is one of the early events to occur. These ROS are either helping in strengthening the cell wall thereby trying to restrict pathogen invasion (*Bradley, Kjellbom & Lamb, 1992*) or act as a signaling molecule to induce defense response such as a rapid HR response (*Alvarez et al., 1998*). Therefore, we suggest that the induced HR response in *atwakl10* might be due to increased pathogen-induced $H_2O_2$ accumulation. We, therefore, sought to determine the accumulation of $H_2O_2$ in WT and KO mutant lines. Our results revealed an increased accumulation of $H_2O_2$ in *atwakl10* mutant line supporting our hypothesis (Fig. 3C).

Promoter analysis of *AtWAKL10* revealed the presence of key TFBS that are involved in abiotic stress tolerance (Fig. 4B). We, therefore, tested *atwakl10* plants for their possible response to salt and drought stress. The *atwakl10* line showed a differential response to both stresses. In the case of salt stress, *atwakl10* plants showed a reduced germination percentage at 150 mM (Figs. 5A and 5B). Literature reports have suggested that cGMP maintains a lower Na ($^+$)/K ($^+$) ratio and increasing the plasma membrane H(+)ATPase gene expression that reduces the injury caused by salt stress (*Li et al., 2011*). *AtWAKL10* that encodes GC and catalyzes the synthesis of cGMP might, therefore, be important in regulating salt stress positively. Contrary to this, in response to drought stress *atwakl10* plants showed a higher survival rate (Figs. 6C and 6D). The promoter of *AtWAKL10* has

TFBS for mostly drought-responsive genes such as *DREB19*, *DREB2* (*Liu et al., 1998*) and ABA-responsive WRKY18 and WRKY40 (Fig. 4B). Previous reports have explored the role of DREBs in the regulation of abiotic stress responses including drought (*Lata & Prasad, 2011*). Similarly, *Chen et al. (2010)* reported the role of WRKY18 and WRKY40 in abscisic acid and abiotic stress responses in Arabidopsis. Transcription factors are regulatory genes that mediate the expression of genes in response to a particular stimulus. Therefore, the presence of TFBS related to drought suggests regulation of *AtWAKL10* by drought-related TFs; however, the exact mechanism needs to be explored further.

## CONCLUSIONS

*AtWAKL10* one of the candidate genes for NO signaling showed 124.7 fold change in response to CysNO in RNA-seq based transcriptomic study (*Hussain et al., 2016*). Investigation into the role of *AtWAKL10* in plant biology using functional genomics revealed that loss of function in *AtWAKL10* resulted in no significant changes in CDF under control and oxidative stress condition whereas under CysNO mediated nitrosative stress condition it showed a significant increase in CDF compared to WT. Furthermore, *AtWAKL10* showed positive regulation of basal and *R*-gene mediated resistance. Promoter analysis of *AtWAKL10* suggested its role in abiotic stresses. We observed that AtWAKL10 positively regulates salt while negatively regulates drought stress. However, further investigation is required for dissecting the underlying pathways that combine or separates biotic and abiotic stress responses.

## ACKNOWLEDGEMENTS

We are thankful to Arabidopsis biological resource center (ABRC) and The European Arabidopsis Stock Center for providing us seeds. We also extend our gratitude to The Arabidopsis information resource (TAIR) for providing useful information about the Gene/Protein function.

### Funding

This research was supported by the National Research Foundation of Korea (NRF) funded by the Ministry of Education Grant Number NRF-2017R1A2B1008820. The funders had no role in study design, data collection and analysis, decision to publish, or preparation of the manuscript.

### Grant Disclosures

The following grant information was disclosed by the authors:
National Research Foundation of Korea.
Ministry of Education: NRF-2017R1A2B1008820.

### Competing Interests

The authors declare that they have no competing interests.

# PeerJ

## Author Contributions

- Phearom Bot performed the experiments.
- Bong-Gyu Mun performed the experiments, analyzed the data, providing experimental tools.
- Qari Muhammad Imran performed the experiments, analyzed the data, prepared figures and/or tables, authored drafts of the paper.
- Adil Hussain conceived and designed the experiments, approved the final draft.
- Sang-Uk Lee contributed in data analysis.
- Gary Loake authored and reviewed drafts of the paper, approved the final draft.
- Byung-Wook Yun conceived and designed the experiments, approved the final draft.

## Data Availability

Raw data are available in the Supplemental Files.

## Supplemental Information

Supplemental information for this article can be found online at http://dx.doi.org/10.7717/peerj.7383#supplemental-information.

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
