# Peer review of "Differential expression of AtWAKL10 in response to nitric oxide suggests a putative role in biotic and abiotic stress responses"

_PeerJ, doi:10.7717/peerj.7383_

## Round 0.1 · original submission · Major Revisions

Both reviewers have rather critical comments. Please consider all the remarks. Additionally, please check English presentation quality. If you need more time for the update - please take it. Despite the critics from the reviewers, welcome to resubmit the manuscript to PeerJ.

Reviewer 1 ·

Basic reporting

The manuscript need to be carefully checked. I could find may typo in the current version.
Suffience background and refs were provided.
The article structure, figure, table and raw data are well prepared.

Experimental design

The authors provided novel findings to provide the role of WAKL10 in biotic and abiotic stress resistance. Experiments are well designed and performed. However, experimental details of how many biological and experimental replicates of experiments were performed need to be provided.

Validity of the findings

The finding of manuscript is novel and important for the understanding of NO responsive gene WAKL10 in biotic and abiotic stress resistance. Although statistical analysis was performed, but some labels are missing the figures. Some of conclusion are not well stated.

Additional comments

In this manuscript, the authors investigated the role of Arabidopsis WAKL10 in biotic and abiotic stress responses. The T-DNA insertion mutant of WAKL10 is insensitive to GySNO treatment, and more susceptible to Pst DC3000, suggesting the dual role of WAKL10 in NO signaling and pathogen resistance. Sufficient background and refs were provided in the manuscript, and the experiments were well performed.
However, the manuscript still need to be improved for publishing in PeerJ.
1. First of all, the manuscript need to be carefully written. I could find quite a lot of typo in the manuscript.
For instance, I was confused with the name of ‘WAKL10’. In the figures, only ‘atwalk’ appears. So, please carefully check and give the full name of WAKL10/WALK10 in the beginning of the manuscript.
Line 139, concentration of ‘5x 105 10 CFU mL-1’, please check.
Pst DC3000 (avrB) should be Pst DC3000 avrB.
‘(Student’s t-test with 95 (*) and 99 % (**) confidence levels. A ‘)’ was lost in the end of the sentence.
… …

2. The authors also investigated the role of atgsnor1-3 for chemical treatment and pathogen treatment. However, why the authors include gsnor1 in this study is unclear, and no further discussion was found for the role of GSNOR in biotic and abiotic stresses.

3. line 216-217. Although PR1 is one of the marker for SA mediated immunity in Arabidopsis. However, the conclusion that ‘due to down regulation of SA-dependent PR genes’ with only PR1 expression may not sufficient.

4. The major point of this manuscript is the role of WAKL10 in ETI. It is reported that HR phenotype, ROS accumulation and iron leakage are accompanied with ETI. The atwakl10 mutant exhibits high level of HR, ROS accumulation and iron leakage, indicating WAKL10 negatively contributes to AvrB triggered ETI in Arabidopsis. This is inconsistent with the bacterial growth results, which makes the authors conclude that WAKL10 is positive to ETI. How would the authors explain this results?

5. Detail information of biological and experimental replicates performed in each experiments need to be described.

6. In the figure legends. The authors mentioned that ‘(Student’s t-test with 95 (*) and 99 % (**) confidence levels). However, some of the ‘*’ was lost in the figures. For instance, Fig. 1C, 2C, 3B, 5B

7. Primer ‘R’ for genotyping need to be provided in table 1.
In addition to this point (but not necessary for this revision), the authors showed that several WRKY and DREB binding site are appeared in the promoter region of WAKL10. Confirmation of WAKL10 expression in wrky and/or dreb mutants under Pst infection or salt treated conditions may provide new evidences to understand how WAKL10 was regulated under biotic and abiotic conditions.

Reviewer 2 ·

Basic reporting

Nitric oxide offers a very important regulatory role in plant response to environmental stresses while NO production is an important feature of plants under biotic and abiotic stresses. In this study an ARABIDOPSIS WALL ASSOCIATED KINASE-LIKE 10 (AtWAKL10) loss-of-function mutant atwakl10 was used to investigate for disease phenotype after Pst infections using both virulent and avirulent strains. Moreover, mutant line was also tested for tolerance against two NO-donors, drought and salinity stress responses regarding germination/survival rates. The gene expression profiling by qPCR also gave interesting data also. However, there are some improvements suggested as below which may be incorporated before the acceptance of this paper.

Throughout the text there is an inconsistent use of WALK10 and WAKL10. Please correct as appropriate. Add some information on atgsnor1-3 and its susceptibility to Pst, since, this mutant has been used in almost all experiments as a positive control with high pathogen growth or HR. There are some languages issues found throughout the text which can be improved with some effort by the authors.

The discussion section is well elaborated. I would suggest to add on a the possible relationship of WALK10 with NO tolerance, and that, why mutant plants showed mortality by adding NO donor (CysNO) in MS media compared to wild-type.

Experimental design

Add in M&M section, what statistics was used. If one-way ANOVA, please mention about F statistics along with number of repeats.

Validity of the findings

The study overall was well conceived and planned. The gene expression profiling, ion-leakage and staining experiments are convincing. The data obtained is of significance in the field of stress biology where a novel AtWAKL10 gene function has been associated with disease resistance and abiotic stress tolerance / susceptibility.

Additional comments

Line 48: Please rephrase. Omit "one of the smallest". "known" not “knows”
Line 51: Please omit miraculous
Line 52: Change citation order throughout the text. Write old ref. first in ascending order. Cite Liu et al., Cell (2004). Essential roles of S-nitrosothiols in vascular homeostasis and endotoxic shock.
Line 53 and 58: add Delledonne et al, 1998 (Nature).
Line 59: please rephrase.
Line 60: Omit atmospheric.
Line 61: replace “may give significant damage” with “cause significant damage”
Line 64: Rephrase “is a key feature in response to”
Line 74: Also cite Jones and Dangl, Nature 2006. The plant immune system.
Rephrase “pathogen recognition obstructs pathogen progression”
Line 84: rephrase “ little is known about its catalytic enzyme GCs”
Line 97: Replace both references with primary references for NO addition to proteins. Add Stamler’s work e.g. PNAS 1992 “S-Nitrosylation of proteins with nitric oxide”
Line 101: Replace “CySNO” with CysNO
Line 102: rephrase “most major”
Line 117 and 130: replace ±23°C with 23°C ± ??
Line 134: Add primary reference.
Line 139: 5 x 105 10 CFU mL looks like a typo. Please mention the inoculum dose for Pst vir and avr, separately.
Line 152: replace “then let to cool down” with” allowed to cool-down.
Line 156: Omit reference. The method described is sufficient. Add OD or cfu at which leaves were infiltrated.
Section 1.6: Mention experimental controls and number of repeats.
Line 170: Omit reference
Line 184: rephrase “Arabidopsis/rice was selected as conserved region”
Line 194 -196: Omit methodology. It is already there in M & M. what is MV?
Line 198: write cell death frequency (CDF) and use abbreviation onward.

Fig. 1(A): Mention in the fig legend, what is F+R and B+R. Omit ladder labelling but bar for the size of the gene band.
Fig. 2(B): Considering almost 2 log difference in pathogen growth for atwakl10 plants compared to wild-type at 2 and 4 dpi depicted by a single asterisk * saying the difference is significant at 95% needs to be rechecked for any possible errors. in my view it could be highly significant at 99% (**).
Fig. 1(D): this figure is not used in the text. Looks like showing morphological attributes of mutant lines compared to Col-0 (wild-type)?

Fig. 2(C): Seems like fold-change in PR relative expression not just relative expression. A high relative PR gene expression at 0 hpi (control) is surprising, since PR gene expression is triggered only after pathogen interaction.

Line 221: replace “within” with “in”
Line 225: Please add significance level (P≤0.01) or (P≤0.05).
Line 231: “posited” ??

Fig. 3(B): A control at day 0 is missing for comparison.

Line 246: Delete reference.
Line 249: Table S1 is missing for commentary.
Line 267: Replace “played” with “play”
Line 269: Italicize DREB2 and 19
Line 271: Please rephrase “the atwakl10 lines were deprived of water for two weeks”
Line 297: Please rephrase “CDF percentage about 80%.”
Line 343: replace “slat” with “salt”
Line 354: delete reference
Line 359: please rephrase “We hope that this article will add to our knowledge about NO biology during plant defense and environmental insults”

---

## Round 0.2 · Major Revisions

This work received some critical comments on the interpretation of the result (conclusion might be misleading). Though review was positive overall, I think in this case it's better solution is to rewrite the work, update it more, change presentation logics.

Reviewer 1 ·

Basic reporting

Professional language checking need to be applied to ensure that international audiences could clearly understand.
References are improved and sufficient information is provided.
The structure of article, figures and tables are clear.

Experimental design

The authors provided novel findings to illustrate the role of AtAWLK10 in plant development, biotic and abiotic stress responses. Sufficient details of methods are described.

Validity of the findings

Comparing with previous version, number of replicate in each experiment and statistical information is provided.

Additional comments

The authors provided interesting finding for the role of AtWALK10 in plant development, biotic and abiotic stress responses, indicating AtWALK10 is important for NO signaling which plays viral role in differential each processes.
However, the major issue of this study is the bacterial growth assay in Figure2 and Figure3. The authors showed in Figure 2 that the Pst population and PR1 expression almost not changed at different time point in Col-0. Similarly, in Figure 3, Pst AvrB growth at 2 and 4 dpi are similar (or even reduced). Those are inconsistent with previous reports. (This may due to unsuccessful infiltration or other factors. However, the authors applied at least three times for each experiments.) Therefore, the conclusion addressed from those experiments might misleading.

Reviewer 2 ·

Basic reporting

This is a revised submission and authors have incorporated almost all the suggestions and managed to clarify many ambiguities.
A few necessary comments have now been incorporated in the main file using Track changes option.

Experimental design

OK

Validity of the findings

OK

Additional comments

A few minor suggestions be addressed.

Annotated reviews are not available for download in order to protect the identity of reviewers who chose to remain anonymous.

---

## Round 0.3 · accepted · Accept

After second review round I see extensive updates in the text covering the remarks from previous review. I believe the manuscript does not need additional review round. So, as an academic editor I recommend accept it in the present form.